

# Fruit and vegetable consumption and anemia among adult non-pregnant women: Ghana Demographic and Health Survey

Bishwajit Ghose[1,2] and  Sanni Yaya[2]

[1] Institute of Nutrition and Food Science, University of Dhaka, Dhaka, Bangladesh
[2] Faculty of Social Sciences, School of International Development and Global Studies, University of Ottawa, Ottawa, Canada

## ABSTRACT

**Background**.  Anemia is the most widely prevalent form of micronutrient deficiency that affects over a quarter of the global population. Evidence suggests that the burden of anemia is higher in the developing countries with women of reproductive age and children being the most at-risk groups. The most common causes are believed to be malnutrition and low bioavailability of micronutrients, which usually result from poor dietary habits and inadequate intake of food rich in micronutrients such as fresh fruits and vegetables. Regular consumption of F&V was shown to have protective effect against NCDs; however, evidence on this protective effect against micronutrient deficiency diseases are limited.

**Objectives**. (1) To measure the prevalence of anemia among adult non-pregnant women in Ghana, and (2) to investigate if there is any cross-sectional relationship between F&V consumption and anemia.

**Methods**. This is a cross-sectional study based on data extracted from the Ghana Demographic and Health Survey, 2008. Subjects were 4,290 non-pregnant women aged between 15 and 49 years. Hemoglobin levels were measured by HemoCue® hemoglobin-meter. Association between anemia and F&V consumption was assessed by multivariable regression methods.

**Results**. Findings indicate that well over half (57.9%) of the women were suffering from anemia of some level. The percentage of women consuming at least five servings of fruits and vegetables a day were 5.4% and 2.5% respectively. Results of multivariable analysis indicated that among urban women, consumption of <5 servings fruits/day was associated with significantly higher odds of severe [AOR = 9.27; 95% CI [5.15–16.70]] and moderate anemia [AOR = 6.63; 95% CI [4.21–10.44]], and consumption of <5 servings of vegetables/day was associated with higher odds of moderate anemia [AOR = 2.39; 95% CI [1.14–5.02]] compared with those who consumed >5 servings/day.

**Conclusion**. The findings indicate that urban women who did not maintain WHO recommended level of F&V consumption bear a significantly higher likelihood of being moderate to severely anemic.

Corresponding author
Bishwajit Ghose,
brammaputram@gmail.com

## INTRODUCTION

Anemia affects more than a quarter of the world's population with most devastating impacts on women of reproductive age (*Rigetti et al., 2012*; *Kassebaum et al., 2014*). A systematic analysis of global anemia burden study reported that with a global prevalence of 32.9%, anemia accounted for 8.8% of the total disability from all conditions in 2010 (*Kassebaum et al., 2014*). While the etiology of anemia is multifactorial, iron deficiency is believed to be the most common underlying cause (*Beck et al., 2014*) as half of the global anemia cases are assumed to be attributable to iron deficiency (*Stoltzfus, 2003*). Several other micronutrients (e.g., vitamins A, B12 or folate) were also found to be important factors in the pathophysiology of anemia, yet these associations remain to be confirmed by interventional studies (*Rigetti et al., 2012*; *Haider & Bhutta, 2011*). Iron deficiency ranks number nine among 26 risk factors included in the Global Burden of Disease (GBD 2000), and accounts for about 841,000 deaths and 35,057,000 disability-adjusted life years lost (*Stoltzfus, 2003*). Although the prevalence and susceptibility of anemia differ substantially across sex and age groups, women of reproductive age are regarded as a research priority because of the increased exposure to pregnancy related complications e.g., preterm labor, placental abruption, low birth weight (LBW), and infant and maternal mortality (*Kassebaum et al., 2014*; *Rasmussen, 2001*; *Ghose et al., 2017*).

Globally, there has been a rising concern regarding the consequences of anemia and other micronutrient deficiency disorders (hidden hunger) on maternal and infant health. Over the course of last two-three decades, significant progress has been achieved towards the elimination of vitamin A and iodine deficiencies, however progress has been uneven toward reducing the burden of iron-deficiency anemia (IDA) across different regions of the world as micronutrient deficiencies continue to be a serious public health issue especially in the South Asia and Sub-Saharan Africa (*Stoltzfus, 2003*; *De Benoist et al., 2008*; *Dalmiya & Schultink, 2003*). According to WHO estimates, the prevalence rates of anemia for non-pregnant women are 47.5% in Africa (highest globally), which makes it of severe public health significance in the region (*De Benoist et al., 2008*).

Deficiencies of certain micronutrients (Iron, Vitamin-A, Iodine) have been a top international health priority for decades, achieving the prevention of which has been considered as one the most important achievable health-related goal (*Maberly et al., 1994*). There is a growing body of research dedicated to investigating the underlying determinants and sustainable interventions of micronutrient deficiencies. The theme that commonly emerge in the current scientific literature is a food-based approach in the context of sustainable agriculture aimed at promoting dietary diversity diversification for long-term intervention of deficiency diseases at population level (*Francesco, Jessica & Emile, 2011*; *Nair, Augustine & Konapur, 2015*). In high anemia prevalent countries e.g., Bangladesh and India, iron supplementation and increased intake of iron rich food have been applied as a strategy to combat anemia among young women. Poor dietary intake and low bioavailability of iron (especially non-heme) in resource-poor countries are considered to be the key determinants of low body iron reserve and resultant anemia (*Jamil et al., 2008*). As dietary factors appear to be the major contributors to iron and other deficiencies

and subsequent development of anemia (*Beck et al., 2014*; *Francesco, Jessica & Emile, 2011*), food-based interventions, such as diet diversification by including seasonal F&V, are usually proposed as a sustainable solution to this persistent problem (*Beck et al., 2014*; *Nair, Augustine & Konapur, 2015*; *Tontisirin, Nantel & Bhattacharjee, 2002*).

The protective mechanism of F&V against non-communicable chronic diseases (NCDs) has been well documented (*Liu et al., 2000*; *Dauchet et al., 2006*; *Hartley et al., 2013*). However, little is known regarding the relationship between F&V intake and anemia or other deficiency diseases. The benefits of F&V consumption in the prevention of NCDs has found to be rooted to numerous phytochemicals they contain which are essential for optimum physiological functioning and prevention of development of metabolic symptoms (*Syed & Yaw Addo, 2016*). For Iron Deficiency Anemia (IDA), F&V exert the beneficial effects mainly through their high non-heme iron content, and through ascorbic acid content (citrus fruits) which contributes to greater bioavailability of iron by functioning as a stimulating factor in its absorption (*Monsen et al., 1978*; *Bothwell et al., 1989*). Though heme iron has higher bioavailability than non-heme iron, the latter occurs in much greater quantity in natural food and thus generally contributes more to iron nutrition than heme-iron food (*Monsen et al., 1978*). Apart from acting as a direct source of dietary iron, F&V consumption exerts the beneficial effects on anemia through their relatively higher vitamin-C content which enhances the bioavailability of iron in the diet. Another crucial micronutrient that makes F&V consumption an important component to consider when it comes to the pathophysiology of anemia is folate (folic acid). Folic acid deficiency-related anemia is a commonly encountered health concern among pregnant women in developing countries, and are routinely advised to take folate or iron-folate supplement during antenatal visits when dietary intake of these elements seem to be insufficient.

Compared to animal based foods, F&V deserve more attention in the context of micronutrient deficiency and dietary diversification especially for their cheaper availability, relatively lower environmental footprints. In addition, F&V has proven benefits in the prevention of several NCDs to which meat consumption has found to be a risk factor (e.g., cancer, diabetes, cardiovascular diseases) (*Genkinger & Koushik, 2007*; *Neal, Susan & Caroline, 2014*; *Kaluza, Wolk & Larsson, 2012*). To this regard, we conducted this study on F&V consumption only to explore whether or not the level of consumption was associated with anemia among non-pregnant women in Ghana. As country representative data on anemia is hard to obtain for developing countries, we used secondary data from the Ghana Demographic and Health Survey (GDHS 2008) which provides information on anemia and F&V consumption among women aged between 15–49 years. The outcomes of this study are expected to contribute to the current literature and future researches by providing insights on how F&V consumption relates to anemia, as well as useful information for anemia prevention strategies among non-pregnant women in Ghana and other countries in the region.

## METHODS

### Data source

Data for this study were obtained from Ghana Demographic and Health Survey (GDHS) that was conducted in 2008. The 2008 GDHS is the fifth of this kind in the country that aimed to provide data to monitor the population and health situation in Ghana under the auspices of Demographic and Health Survey of USAID. The survey included a range of demographic, socioeconomic, health knowledge and behavior related themes. The survey was carried out by the Ghana Statistical Service (GSS) and the Ghana Health Service (GHS) with technical assistance of ICF Macro, (an ICF International Company) through the MEASURE DHS program and financial support from the United States Agency for International Development (USAID). The fieldwork lasted from September 8 to November 25 of the year 2008. The survey included three different types questionnaires: for Household, Women, and Men. The dataset for the present study was from the questionnaire for women.

### Sampling strategy

The 2008 GDHS was a household-based survey, implemented in a representative probability sample of more than 12,000 households from urban and rural areas in each of the 10 regions in Ghana. A two-stage sampling design was utilised. The first stage involved selection of cluster from a master sampling frame, and the second stage involved selection of households from the clusters. For clusters selection, a master sampling frame was used that was designed during the 2000 Ghana Population and Housing Census. In total 412 clusters were selected from the master sampling frame. In the second stage, 30 households were selected systematically from each cluster. Weights were also calculated taking into consideration the non-responses at various level to ensure the sample is representative of the general population. Of the 12,323 households selected in the sample, 11,913 were occupied at the time of the fieldwork. A total of 5,096 eligible women were identified in the households and were selected for individual interview. Finally, interviews were successfully completed with 4,916 of the women with a response rate of 97%. More details regarding the sampling techniques are available elsewhere (*GSS, GHS & ICF Macro, 2009*).

### Hemoglobin testing for anemia

For GDHS 2008, diagnosis of anemia was performed by using the HemoCue® blood hemoglobin testing system. Hemoglobin testing is the routinely used method for anemia diagnosis. HemoCue® (HemoCue Inc., Mission Viejo, CA, USA) is a user friendly and highly reliable point-of-care testing (POCT) system and one of the most commonly utilized of hemoglobin testing devices (*Ghose et al., 2016a*). The protocol for hemoglobin testing was approved by the ICF Macro Institutional Review Board (IRB) in Calverton, Maryland, USA and the Ghana Health Service Ethical Review Committee in Accra, Ghana. Trained surveyors were responsible for undertaking the HemoCue® test through finger prick method. Women from half of the households who participated in the 2008 GDHS survey were selected (those who gave consent) for the anemia test.

## Variables selection

Outcome variable: Anemia status (measured in terms of Hb concentration) was the outcome variable in this study. Anemia was categorized as per the cut-off points recommended by the WHO for women above age 15 years: Mild anemia = 10–11.9 g/dl; moderate anemia = 7–9.9 g/dl; severe anemia = <7.0 g/dl, non-anemic = ≥12 g/dl (*Ghose et al., 2016b*).

## Independent variables

Levels of fruit and vegetable consumption were the main independent variables in this study. Respondents were asked separately how many servings of fruits and vegetables they consume on a typical day. For both fruits and vegetables, the answers were categorized according to WHO recommendations for adult individuals as: ≥5 servings of fruits/day = adequate, and <5 servings of fruits/day = inadequate.

Guided by existing literature on dietary intervention and determinants of anemia/micronutrient deficiency and availability on the datasets, we have considered several socioeconomic, demographic and health behaviour related variables as potential confounders, and were categorized as: **Age**: 15–19, 20–24, 25–29, 30–34, 35–39, 40–44, 45–49; **Marital status:** Never married, Married, Living together, Other; Religion: Christian, Moslem, Other; **Educational attainment**: No education, Primary, Secondary, Higher; **Wealth status:** Poorest, Poorer, Middle, Richer, Richest: **BMI**: Underweight, Normal weight, Overweight, Obese (*Nair, Augustine & Konapur, 2015*; *Tontisirin, Nantel & Bhattacharjee, 2002*; *Liu et al., 2000*; *Monsen et al., 1978*; *GSS, GHS & ICF Macro, 2009*; *Ghose et al., 2016a*; *Ghose et al., 2016b*; *Bishwajit, Sanni & Shangfeng, 2016*). Body mass index (BMI) was defined as weight (kg) divided by height (m$^2$), and was categorized as per WHO definitions: <18.49 = Underweight, 18.5–24.9 = Normal weight, 25–29.9 = Overweight, >30 = Obese. Wealth index was included in the study as it is used universally in DHS studies as a proxy indicator for economic status of households that is consistent with expenditure and income measures (*GSS, GHS & ICF Macro, 2009*). The calculation of the wealth index involves assigning a factor score for a set of household possessions (e.g., TV, radio, bicycle, and housing quality e.g., type of floor, wall, and roof) which is generated through principal component analysis (PCA). The scores are then summed and standardized for each household which places them in a continuous scale based on the individual scores. Once calculated, the scores are then categorized into quintiles where each household falls into a category, with the lowest scores representing the poorest and highest representing the richest households (*Ghose et al., 2017*).

## Statistical analysis

All analyses were conducted with SPSS 21 for Mac (SPSS Inc., Chicago, IL, USA). Initial checks were performed for missing values, outliers and multicollinearity issues. Sample weight was applied (as provided in the dataset) before analysing the dataset to account for any variations in the response rates across survey regions. Results were stratified by place of residency, as people in urban and rural areas have been shown to differ significantly in terms of exposure various health risk factors, lifestyle and health-related

behaviours (*SE & Elsie, 2004*). Basic sample characteristics (Socioeconomic, demographic, behavioural) were presented as frequencies and percentages. Pearson's Chi-square tests were performed to check for statistical association between anemic and non-anemic groups and F&V consumption status plus all other covariates. Only those variables that were found significant in Chi-square tests at $p < 0.25$ were included in the multivariable regression model. The results of regression analysis were presented as odds ratios and corresponding 95% confidence intervals (CI). All tests were two-tailed, and statistical significance was set at $p < 0.05$.

### Ethics approval

All participants gave informed consent before taking part in the interview. The ICF International was responsible for ensuring that the survey complies with the US Department of Health and Human Services regulations for the protection of human subjects, and the host country ensures that the survey complies with laws and norms of the nation (*GSS, GHS & ICF Macro, 2009*). Further approval for using DHS datasets are unnecessary since they are available in the public domain without any identifiable information of the participants.

## RESULTS

### Profile of the respondents

Basic demographic and socioeconomic characteristics of the sample population were presented in Table 1. Mean age of the participants were 29.3 years (SD 9.8). Both in urban and rural areas, majority of the women were aged between 15 to 29 years. Less than half of the women were married (45.6%), and almost four-fifth belonged to Christian faith (78.5%). One-fifth of the women had no formal education (20.6%), and majority had secondary level qualification (55.5%). Women in urban areas had higher rates of secondary (67.2% Vs 44.4%) and higher than secondary (6.1% Vs 1.1%) level educational qualification. About one-third of the women reported living in poorest to poorer households (33.6%), while 22.8% were living in households with highest wealth status. Regarding body weight status, 8.4% of the women were underweight, while the rate of overweight and obesity were respectively 20% and 10%. Chi-square tests also revealed significant urban-rural disparities in all the demographic and socioeconomic variables listed in Table 1, whereas for the behavioral factors the difference was significant for fruit consumption only (Table 2).

### Prevalence of health-related behaviours (smoking, drinking, F&V intake) and anemia

The prevalence of alcohol consumption was 18.1%, while that of tobacco smoking was almost nil (0.1%). Regarding fruit and vegetable intake, only 5.4% and 2.5% women reported consuming more than five servings of fruits and vegetables respectively. Significantly different prevalence was observed between urban and rural women in terms of adequate consumption (5 or >5 servings/day) of fruits, but not vegetables. The prevalence of anemia was considerably high as more than half of the women (57.9%) had anemia of some level. The prevalence of severe, moderate and mild anemia was respectively 1.6%, 16.1% and 40.2% (Table 2). Women in rural areas had higher prevalence of anemia of all three levels compared with their urban counterparts.

**Table 1  Participant characteristics (Socioeconomic and demographic).** GDHS 2008.

| Variables | Total (4,290) | Urban (2,090) | Rural (2,200) | P |
|---|---|---|---|---|
| **Age** | 29.3 (SD 9.8) | 28.62 (SD 9.6) | 29.42 (SD 9.8) | |
| 15–19 | 933 (21.7) | 459 (22.0) | 474 (21.5) | |
| 20–24 | 734 (17.1) | 388 (18.6) | 346 (15.7) | |
| 25–29 | 704 (16.4) | 347 (16.6) | 357 (16.2) | 0.019 |
| 30–34 | 532 (12.4) | 272 (13.0) | 260 (11.8) | |
| 35–39 | 572 (13.3) | 265 (12.7) | 307 (14.0) | |
| 40–44 | 427 (9.9) | 186 (8.9) | 241 (10.9) | |
| 45–49 | 387 (9.0) | 172 (8.2) | 215 (9.8) | |
| **Marital status** | | | | |
| Never married | 1,440 (33.6) | 754 (36.1) | 686 (31.2) | |
| Married | 1956 (45.6) | 910 (43.5) | 1,046 (47.5) | 0.004 |
| Living together | 492 (11.5) | 225 (10.7) | 267 (12.1) | |
| Other | 403 (9.4) | 44 (2.1) | 201 (9.1) | |
| **Religion** | | | | |
| Christian | 3,370 (78.5) | 1,680 (80.4) | 1,689 (76.8) | |
| Moslem | 613 (14.3) | 364 (17.4) | 249 (11.3) | <0.001 |
| Other | 307 (7.2) | 45 (2.2) | 262 (11.9) | |
| **Educational attainment** | | | | |
| No education | 883 (20.6) | 222 (10.6) | 661 (30.0) | |
| Primary | 875 (20.4) | 336 (16.1) | 539 (24.5) | <0.001 |
| Secondary | 2,382 (55.5) | 1,405 (67.2) | 977 (44.4) | |
| Higher | 151 (3.5) | 127 (6.1) | 24 (1.1) | |
| **Wealth status** | | | | |
| Poorest | 666 (15.5) | 16 (0.8) | 650 (29.5) | |
| Poorer | 775 (18.1) | 93 (4.5) | 682 (31.0) | |
| Middle | 874 (20.4) | 376 (18.0) | 497 (22.6) | <0.001 |
| Richer | 999 (23.3) | 721 (34.5) | 277 (12.6) | |
| Richest | 977 (22.8) | 883 (42.2) | 94 (4.3) | |
| **BMI** | | | | |
| Underweight | 360 (8.4) | 125 (6.0) | 235 (10.7) | |
| Normal weight | 2,643 (61.6) | 1,123 (53.7) | 1,520 (69.1) | <0.001 |
| Overweight | 859 (20.0) | 531 (25.4) | 329 (14.9) | |
| Obese | 427 (10.0) | 312 (14.9) | 116 (5.3) | |

**Notes.**
N.B. *p*-values (from Chi-square tests) indicate the significance in the differences between urban and rural sample for the explanatory variables.

## Bivariate association between anemia and the socioeconomic, demographic and health behavioral characteristics

We examined bivariate differentials to explore the differences in anemia prevalence across the categories of the explanatory variables. Table 3 indicates that the prevalence of severe anemia was highest among women aged 45–49 years, while that of moderate and mild anemia was highest among those in the age group of 15–19 years. No significant

**Table 2  Health-related behavior and body weight status of the sample population.** GDHS 2008.

| Variables | Total | Urban | Rural | *p*-value |
|---|---|---|---|---|
| **Alcohol consumption** | | | | |
| No | 3,514 (81.9) | 1,715 (82.1) | 1,799 (81.7) | 0.408 |
| Yes | 776 (18.1) | 375 (17.9) | 402 (18.3) | |
| **Smokes cigarettes** | | | | |
| No | 4,286 (99.9) | 2,087 (99.9) | 2,199 (99.9) | 0.293 |
| Yes | 4 (0.1) | 3 (0.1) | 1 (0.1) | |
| **Fruit consumption** | | | | |
| <5 servings/day | 4,057 (94.6) | 1,950 (93.3) | 2,107 (95.8) | **<0.001** |
| 5 or >5 servings/day | 233 (5.4) | 140 (6.7) | 93 (4.2) | |
| **Vegetable consumption** | | | | |
| <5 servings/day | 4,184 (97.5) | 2,042 (97.7) | 2,142 (97.3) | 0.269 |
| 5 or >5 servings/day | 106 (2.5) | 48 (2.3) | 58 (2.7) | |
| **Anemia level** | | | | |
| Severe | 68 (1.6) | 27 (1.3) | 41 (1.9) | |
| Moderate | 691 (16.1) | 332 (15.9) | 360 (16.3) | **0.001** |
| Mild | 1,723 (40.2) | 789 (37.8) | 934 (42.4) | |
| Not anemic | 1,808 (42.1) | 942 (45.1) | 866 (39.3) | |

**Notes.**
N.B. *p*-values indicate the significance in the differences between urban and rural sample for the individual behavioral factors.

difference was found between anemia and marital status and religion. However; educational attainment ($p = 0.001$), household wealth status ($p = 0.04$), and BMI ($p < 0.001$) appeared to be significantly associated with anemia. Table 3 also indicates that the likelihood of being anemic (all three levels) was higher among those who had secondary level education, were from middle wealth status households, and were underweight. Regarding the health behavior related characteristics, significant association was found between anemia with fruits and vegetable consumption, but not with smoking tobacco and drinking alcohol (Table 4).

## Multivariable analysis

Results of multivariable analysis for measuring the association between F&V consumption and mild, moderate and severe anemia among urban and rural women are shown in Table 5. Results indicate that inadequate fruit consumption was associated with higher odds of severe and moderate anemia, but not mild anemia among urban women. No statistically significant association was observed between fruits and vegetable consumption and anemia among rural women. Compared with women who consumed at least five servings of fruits per day, the odds of being severely anemic was over nine times [AOR = 9.28; 95% CI [5.15–16.70]] and of being moderately anemic was over six times [AOR = 6.64; 95% CI [4.21–10.44]] among those who reported consuming less than five servings of fruits a day. Regarding vegetable consumption, women who did not consume at least five servings a day had 2.4 [AOR = 2.39; 95% CI [1.14–5.02]] times higher odds of being moderately anemic compared to those who consumed five servings of fruits per day.

**Table 3** **Proportion of sample population with mild, moderate and severe anemia across the socioeconomic and demographic characteristics.** GDHS 2008.

| Variables | Severe | Moderate | Mild | p |
|---|---|---|---|---|
| **Age** | | | | |
| 15–19 | 8.8 | 22.3 | 24.2 | |
| 20–24 | 16.2 | 16.2 | 17.4 | |
| 25–29 | 10.3 | 16.8 | 14.9 | <0.001 |
| 30–34 | 5.9 | 10.3 | 13.1 | |
| 35–39 | 17.6 | 15.8 | 13.1 | |
| 40–44 | 16.2 | 10.9 | 9.8 | |
| 45–49 | 25.0 | 7.8 | 7.7 | |
| **Marital status** | | | | |
| Never married | 34.8 | 35.4 | 33.2 | |
| Married | 47.8 | 41.6 | 45.9 | 0.690 |
| Living together | 10.1 | 12.7 | 11.3 | |
| Other | 7.2 | 10.3 | 9.6 | |
| **Religion** | | | | |
| Christian | 80.6 | 76.7 | 78.6 | |
| Moslem | 14.9 | 16.4 | 14.0 | 0.684 |
| Other | 4.5 | 6.9 | 7.4 | |
| **Educational attainment** | | | | |
| No education | 17.6 | 22.9 | 19.9 | |
| Primary | 27.9 | 23.3 | 21.4 | 0.001 |
| Secondary | 52.9 | 50.5 | 56.1 | |
| Higher | 1.5 | 3.3 | 2.6 | |
| **Wealth status** | | | | |
| Poorest | 18.8 | 15.5 | 16.1 | |
| Poorer | 23.2 | 17.5 | 19.9 | |
| Middle | 24.6 | 24.3 | 22.8 | 0.04 |
| Richer | 21.7 | 21.7 | 20.1 | |
| Richest | 11.6 | 21.1 | 21.2 | |
| **BMI** | | | | |
| Underweight | 61.2 | 59.1 | 65.5 | |
| Normal weight | 26.9 | 21.2 | 16.9 | <0.001 |
| Overweight | 10.4 | 11.4 | 8.1 | |
| Obese | 1.5 | 8.2 | 9.5 | |

**Notes.**
N.B. *p* calculated from chi-square tests.

# DISCUSSION

The aim of the present study was to measure the prevalence of anemia, and its association with fruits and vegetables consumption in a nationally representative non-pregnant women sample in Ghana. Results indicated that over half of the women were living with anemia of some degree (Severe: 1.6%, Moderate: 16.1%, Mild: 40.2%), which is higher than average for African countries (57.9% Vs 47.5%) (*De Benoist et al., 2008*). According to

**Table 4  Proportion of sample population with mild, moderate and severe anemia across the behavioral characteristics.** GDHS 2008.

|  | Severe | Moderate | Mild | $p$ |
|---|---|---|---|---|
| **Alcohol consumption** | | | | |
| No | 83.8 | 82.6 | 82.9 | 0.283 |
| Yes | 16.2 | 17.4 | 17.1 | |
| **Smokes cigarettes** | | | | |
| No | 100.0 | 100.0 | 99.8 | 0.834 |
| Yes | 0 | 0 | 0.2 | |
| **Fruit consumption** | | | | |
| <5 servings/day | 72.1 | 88.1 | 96.6 | **<0.001** |
| 5 or >5 servings/day | 27.9 | 11.9 | 3.4 | |
| **Vegetable consumption** | | | | |
| <5 servings/day | 95.6 | 96.1 | 97.7 | **<0.001** |
| 5 or >5 servings/day | 4.4 | 3.9 | 2.3 | |

**Notes.**
  N.B. $p$ calculated from chi-square tests.

WHO classifications, at this prevalence rate anemia qualifies to be of severe public health significance in the country (*De Benoist et al., 2008*). Not surprisingly, the prevalence rates of all three levels of anemia in the rural areas were higher than in urban areas. Regarding F&V intake, it appeared that most of the women did not maintain adequate amount (in line with WHO recommendation) of consumption as only 5.4% and 2.5% women reported consuming at least five servings of fruits and vegetables respectively. Percentage of women who consumed at least five servings of fruits was higher in urban areas, whereas that of vegetable was slightly higher in rural areas. Our findings also indicated that compared to women who consumed at least five servings of fruits and vegetables, those who consumed less than five servings of fruits and vegetables had higher odds of suffering from severe and moderate anemia and moderately anemia respectively. However, the association was statistically significant for urban women only. It is hard to explain this urban-rural variation regarding this association within the scope of the current study. It is assumable the etiology of might be as well linked to factors other than dietary ones such as parasitic infestations or inflammatory diseases to which rural population are particularly vulnerable (*Ponder & Long, 2013*).

Despite its significant public health importance, research evidence on anemia and the associated risk factors remain scarce in Ghana. No comparable nationwide estimates are available as yet for non-pregnant women. A cross-sectional study among pregnant women based on Accra found high rates anemia (All cause anemia = 34%, Iron deficiency anemia = 7.5%) (*Engmanna et al., 2008*). Although evidence from countries in sub-Saharan Africa is scarce, similarly high prevalence rates were reported in South Asian countries e.g., India (mild anemia = 32.4%, moderate anemia = 14.19%, severe anemia = 2.2%) (*Bentley & Griffiths, 2003*), and in Bangladesh (mild anemia = 35.5%, moderate anemia = 5.6%, severe anemia = 0.2%) which are recognised as global hotspots for micronutrient deficiency diseases (*Kamruzzaman et al., 2015*). This study

**Table 5  Multivariate association between fruit and vegetable consumption and different levels of anemia among non-pregnant women in Ghana.** GDHS 2008.

| | Severe | | Moderate | | Mild | |
|---|---|---|---|---|---|---|
| | POR (95% CI) | AOR (95% CI) | POR (95% CI) | AOR (95% CI) | POR (95% CI) | AOR (95% CI) |
| **(Urban sample)** | | | | | | |
| **Fruit consumption** | | | | | | |
| ≥5 servings/day | | ref | | ref | | ref |
| <5 servings/day | 8.872 | **9.279** | 5.216 | **6.639** | 1.222 | 1.017 |
| | (2.381–33.066) | **(5.155–16.703)** | (1.420–19.162) | **(4.219–10.447)** | (0.279–5.356) | (0.607–1.705) |
| **Vegetable consumption** | | | | | | |
| ≥5 servings/day | | ref | | ref | | ref |
| <5 servings/day | 5.308 | 3.708 | 2.007 | **2.386** | 1.765 | 0.910 |
| | (1.542–18.271) | (0.544–25.290) | (0.664–6.070) | **(1.135–5.018)** | (0.554–5.624) | (0.437–1.896) |
| **F&V consumption** | | | | | | |
| ≥5 servings/day | | ref | | ref | | ref |
| <5 servings/day | 1.677 | 1.011 | 1.684 | 0.867 | 0.605 | 0.971 |
| | (0.594–4.735) | (0.570–1.793) | (0.663–4.275) | (0.708–1.060) | (0.146–2.507) | (0.831–1.134) |
| **(Rural sample)** | | | | | | |
| **Fruit consumption** | | | | | | |
| ≥5 servings/day | | ref | | ref | | ref |
| <5 servings/day | 1.742 | 1.242 | 1.832 | 1.265 | 0.706 | 0.636 |
| | (0.419–7.250) | (0.338–4.563) | (0.201–16.660) | (0.712–2.246) | (0.280–1.782) | (0.386–1.048) |
| **Vegetable consumption** | | | | | | |
| ≥5 servings/day | | ref | | ref | | ref |
| <5 servings/day | 1.962 | 1.409 | 1.607 | 1.236 | 1.674 | 1.301 |
| | (1.060–3.635) | (0.211–9.398) | (0.887–2.909) | (0.571–2.678) | (0.900–3.112) | (0.700–2.417) |
| **F&V consumption** | | | | | | |
| ≥5 servings/day | | ref | | ref | | ref |
| <5 servings/day | 1.501 | 1.343 | 1.958 | 0.880 | 1.138 | 1.054 |
| | (0.806–2.796) | (0.636–2.834) | (0.517–7.418) | (0.673–1.152) | (0.841–1.541) | (0.857–1.296) |

Notes.

N.B. Significant ORs were shown in bold.

POR, partially adjusted odds ratio (adjusted for age, educational attainment, wealth status); AOR, Adjusted odds ratio (adjusted for age, educational attainment, wealth status, BMI).

recognizes lack of prior researches on the association between fruit and vegetable consumption and anemia among women. Only a recent Brazilian study on children aged 4–10 years reported that children who consumed lower than usual amount of F&V were twice as likely to have micronutrient deficiencies compared to children with usual F&V consumption (*Augusto et al., 2015*). The reason why this topic has failed to receive adequate research attention is probably because of the absence of any universal consensus on the level of F&V intake for optimum nutrition. Another inherent complexity in assessing the impact of F&V consumption on micronutrient deficiency outcomes is inadequate knowledge base regarding the dietary and physiological factors limiting and stimulating absorption of certain types of micronutrients (*Beck et al., 2014*).

Besides the poor level of F&V consumption, there were also indications of urban-rural differences in the patterns of F&V intake. The noticeably lower intake of F&V might be because of the nutrition transition the country is undergoing (*Bishwajit et al., 2014*), which is being led by rapid urbanization, improving living standards and changing food policy dynamics (*Ghose, 2015*; *Agyei-Mensah & Aikins, 2010*). The shift in dietary habits such as increased intake of convenience foods and lower intake of fresh F&V have raised concerns regarding the nutritional status of the population in both developing and developed countries (*Bishwajit et al., 2014*; *Agyei-Mensah & Aikins, 2010*). Though the present study does not provide any insight on dietary preferences among the participants, the dietary transition and low F&V consumption (*Bishwajit et al., 2014*) might offer some explanation on the high prevalence of anemia among the participants.

The public-health importance of addressing micronutrient deficiencies has been underscored by major international nutrition conferences and agencies who have played critical roles in advocating for and raising awareness among policymakers and the general population regarding these issues at the global, regional, and national levels (*Haider & Bhutta, 2011*; *Dalmiya & Schultink, 2003*). There is a growing body of research confirming that the underlying determinants of micronutrient deficiencies, and a sustainable solution to this persistent problem are best understood by taking into consideration the complex interplay of health, agricultural and sociopolitical factors. Dietary diversification especially by inclusion of seasonal fruits and vegetables, as opposed to supplementation and fortification, has been proposed to be a more culturally adaptable and a sustainable strategy that hold of the potential to address multiple micronutrient deficiencies simultaneously (*Tontisirin, Nantel & Bhattacharjee, 2002*). Findings of this study support the consensus that optimum consumption of F&V might prove beneficial for addressing the burden of anemia among Ghanaian women.

As far as we are concerned, this is the first study to report on F&V consumption in association with anemia among non-pregnant women. Sample size was considerably large and the findings are generalizable for non-pregnant women population aged between 15–49 years. Diagnosis and categorization of anemia was based on standard procedures. Data were analysed with rigorous methods and reported as per STROBE guidelines. Despite the strengths, there are some important limitations that needs to be considered when interpreting the findings. Firstly, the data were secondary which means that we could not control the measurement and selection of variables and confounders. The cross-sectional nature of the data precludes making any causality regarding the relationship between outcome and explanatory variables. Therefore, the association between fruit and vegetable consumption and anemia provides no indication of direction, which may be bidirectional. It is possible the anemia affects taste acuity/appetite which might reduce the intake of fruits and vegetables. We were also limited in our ability to determine the types of F&V that were consumed (diversity in F&V intake). This is an important concern since the micronutrient content of F&V varies considerably by type, and across seasons and regions. As fruit and vegetables consumption was self-reported, there remains the chance of over/underreporting. Last but not least, we could not adjust the analysis for the presence of non-dietary

factors that are known contributing factors to anemia (e.g., problems with absorption, parasitic infestation, drug toxicity or inflammatory diseases) which might have caused any misinterpretation of the prevalence of anemia.

## CONCLUSION

This study concludes that the overall prevalence of anemia among non-pregnant women in Ghana was higher than that of the African country average. Based on the WHO classification, the prevalence is categorized as a severe public health concern. Our study also found a positive association between F&V consumption and anemia among non-pregnant women living in the urban areas. Results of this study might be of crucial importance for public health programs targeting anemia prevention in the country. From the findings, it is also recommendable that national nutrition programs focus on developing strategies to promote F&V consumption as a part of the policy to address anemia among adult non-pregnant women, especially in urban areas. To provide more precise estimates of national anemia prevalence, future studies should take into consideration other disease conditions that might have affected the prevalence of anemia.

### List of abbreviations

| | |
|---|---|
| **GDHS** | Ghana Demographic and Health Survey |
| **NCDs** | Non-communicable chronic diseases |
| **IDA** | Iron-deficiency anemia |
| **MDGs** | Millennium Development Goals |

### Funding

The authors received no funding for this work.

### Competing Interests

The authors declare there are no competing interests.

### Author Contributions

- Bishwajit Ghose conceived and designed the experiments, performed the experiments, analyzed the data, prepared figures and/or tables, authored or reviewed drafts of the paper.
- Sanni Yaya analyzed the data, contributed reagents/materials/analysis tools, prepared figures and/or tables, authored or reviewed drafts of the paper.

### Human Ethics

The following information was supplied relating to ethical approvals (i.e., approving body and any reference numbers):

The dataset is secondary and available in the public domain in anonymized form for open research. The survey was approved by ICF Institutional Review Board which is a global program of USAID.

https://dhsprogram.com/What-We-Do/Protecting-the-Privacy-of-DHS-Survey-Respondents.cfm.

## Data Availability

The raw data is provided in a Supplemental File.

## Supplemental Information

Supplemental information for this article can be found online at http://dx.doi.org/10.7717/peerj.4414#supplemental-information.

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
