# Peer review of "Fruit and vegetable consumption and anemia among adult non-pregnant women: Ghana Demographic and Health Survey"

_PeerJ, doi:10.7717/peerj.4414_

## Round 0.1 · original submission · Major Revisions

Dear Bishwajit,

I believe that your paper is a good contribution to the field. After reading over the comments from reviewer 1, I agree with them. Please address each specific point made by the reviewer and also work on an English editing of your paper to increase clarity.

·

Basic reporting

1. I would like the introduction to make it clear why the particular hypothesis being tested (that there is an association between fruit or vegetable intake and anemia) is worth examination. It seems that a more general exploratory exercise of examining which characteristics are associated with anemia in this population might have been more easily justified than specifically looking at fruit and vegetable consumption (in the absence of any other dietary data) on anemia. Then the purpose of the study would be to examine where the burden of anemia lies (i.e.: with whom) and behavioural correlates. Because you have no other components of diet, it is very difficult to be sure that any association between fruit or vegetable consumption and anemia is actually specific to this aspect of the diet, or whether fruit or vegetable consumption are just markers of diet quality (and perhaps those eating more fruit are also eating more meat, a more varied diet, or more food in general).

2. Throughout you refer to fruit and vegetable consumption. But you actually looked at fruit consumption and vegetable consumption separately so I think much of your manuscript needs you to address this by changing "fruit AND vegetable consumption" to "fruit OR vegetable consumption"

3. The introduction needs attention in order to improve the English. Specifically, the grammar is confusing.

4. I believe Tables should be standalone- ie: In Table 1 you need to state what the p-values are referring to, either in the table or in the legend. In Table 5, you need to make it clear in the table or legend that the first row is urban women and the second row is rural women (as can be deduced from the text). (Otherwise my assumption would be that the first row was reporting an unadjusted model and the second row was reporting an adjusted model).

5. Line 66 should read "increased exposure *to* pregnancy related complications".

Experimental design

This is an original primary research article within the Aims and Scope of the journal.

The research question needs to be better defined and justified in the introduction (see above).

The methods require more detail and some decisions are not justified.
1. I assume that "a portion" was defined as 80g in order to match the WHO recommendation (ie: of 400g per day)? Please state this clearly if so. Were women told how to estimate a portion, or did they give natural portions and these were converted into 80g portions later?
2. I am not sure why fruit intake and vegetable intake have been examined separately- is there justification for this (e.g.: evidence of a differential effect)? Note that the WHO recommends 400g of fruit *and* vegetables- so describing women's intake as inadequate based on just one of these needs further explanation. If possible a variable which described <5 and >5 portions of fruit and vegetables would be more meaningful (unless there is justification for examining these separately).
3. Given that the DHS collects data on pregnancies/births- I am surprised that you didn't consider this an important variable to examine with respect to risk of anemia (given as you say in the introduction that child-bearing is one of the reasons that anemia is important in women). Parity could have been included as a variable in the multivariate analysis?

Validity of the findings

1. Findings must be more carefully described
eg1: "the majority of women were aged between 15 and 19" (line 218). This is not true. 21.7% of the population were in that age category according to Table 1, so you can say "the modal age category was 15-19" or "the majority of women were aged 15-29" (because more than 50% of your sample are in those 3 age-categories).

eg2: "considerable difference in participation was observed from poorest and richest households between urban and rural women" (line 227-228). I don't think this is accurate. You are not reporting participation rates here. Do you mean that the richest quintile are over-represented (probably because they are more likely to live) in urban areas, while the poorest quintile are over-represented (probably because they are more likely to live) in rural areas (that is how I interpret Table 1). By using the word participation it suggests that in urban areas poorer women were less likely to participate- but I don't think that is what you mean.

2. All of the variables examined were significantly different between Urban and Rural groups. This could justify why you have stratified your later analyses by place. I think this should be explicitly stated (instead you have only commented on wealth and BMI as significantly different between urban and rural groups).

3. In your multivariate analysis I think it would be useful to see unadjusted, adjusted (as you have presented) and fully adjusted models (including all the variables you considered as possible confounders- because although smoking may not have come out as associated in bivariate analyses, might this be because it was confounded by wealth status in these analyses, but it may still be important in multivariate analysis). I am particularly interested in comparing the unadjusted model to the model after adjusting for BMI (see below...).

4. I feel strongly that the most obvious explanation for your findings is that F/V intake is simply a signal for dietary quality and that this makes it more likely that there is something else about the diet that is the real reason for differential anemia prevalence. I think examining the OR for F+V and for BMI with anemia might suggest that diet quality is associated with anemia and that is a more justifiable conclusion that the one you currently draw. I am keen to know whether the adjustment for BMI had a large effect on the OR for fruit and/or veg consumption and anemia as I think you are more likely seeing an effect of diet quality on anemia.

Additional comments

It is not clear why this particular hypothesis is being tested from your current introduction. Given that this is your hypothesis, I think a variable that looks at fruit AND vegetable consumption is really important.

I look forward to seeing the revised manuscript in due course.

---

## Round 0.2 · accepted · Accept

Thank you for your rebuttal letter and for addressing the points raised by the prior reviewer. I believe that you and your team have done a good job responding to each of the concerns and your paper is now acceptable for publication.